**∂** | **Open Peer Review** | Clinical Microbiology | Research Article

# Precise species identification and whole-genome sequencing analysis of *Enterobacter cloacae* complex causing bloodstream infections in China

Yanbing Li,[1,2] Ziran Wang,[1] Ge Zhang,[1] Wei Kang,[1] Jin Li,[1] Yingchun Xu,[1] Menglan Zhou[1]

**ABSTRACT**   The *Enterobacter cloacae* complex (ECC) is one of the major causes of hospital-acquired infections. However, achieving accurate species identification and comprehensive resistance profiling remains difficult in clinical practice, with a limited understanding of species-specific resistance patterns. ECC isolates were collected from patients with bloodstream infections at Peking Union Medical College Hospital between 2015 and 2020. Whole-genome sequencing (WGS) was performed to identify species, analyze antimicrobial resistance genes, and explore genomic variation in serial isolates. Multi-locus sequence typing (MLST) profiles were extracted from the WGS data. Phylogenetic analysis was conducted based on *hsp60* sequences. Eleven *hsp60* clusters were identified, with cluster VIII being the most prevalent (28/108). Average nucleotide identity (ANI)-based species classification showed *Enterobacter hormaechei* (31.5%) and *Enterobacter xiangfangensis* (15.7%) were dominant species. Five clade-cluster pairs (B-VIII, A-VI, G-XI, D-III, R-IX) accounted for 74% of isolates. A total of 90 sequence types (STs) were detected, including 29 novel STs. Resistance gene analysis revealed a high prevalence of $bla_{ACT}$, with distinct distribution patterns observed among different species. Twenty isolates were carbapenem-resistant, with three carrying $bla_{NDM-1/5}$. *Enterobacter roggenkampii* was the most common species (5/20) among all carbapenem-resistant isolates, and 83.3% of the isolates showed resistance to both carbapenems and colistin. Comparative genomics of longitudinal isolates from individual patients revealed adaptive single-nucleotide polymorphisms (SNPs) in *pco* genes. This study provides a detailed genomic characterization of ECC isolates from bloodstream infections, highlighting species diversity, resistance gene distribution, and potential within-host evolution. These insights advocate genome-based surveillance in managing ECC infections and understanding resistance evolution in clinical contexts.

**IMPORTANCE**   *Enterobacter cloacae* complex (ECC) is a major cause of hospital-acquired bloodstream infections, yet species-level identification and resistance profiling remain challenging. As one of the largest whole-genome sequencing (WGS)-based studies of ECC bloodstream isolates in northern China to date, we performed whole-genome sequencing of 108 ECC isolates, revealing high genetic diversity and identifying 29 novel sequence types. We clarified the correspondence between species, clades, and clusters and highlighted *Enterobacter roggenkampii* as a potential high-risk species linked to carbapenem and colistin resistance. Our findings not only improve the understanding of ECC population structure and resistance evolution in China but also provide valuable genomic data for future epidemiological surveillance and species-level diagnostics.

**KEYWORDS**   *Enterobacter cloacae* complex, bloodstream infection, whole-genome sequencing

Address correspondence to Menglan Zhou, mumuxi529@139.com, or Yingchun Xu, xycpumch@139.com.

The authors declare no conflict of interest.

See the funding table on p. 12.

The *Enterobacter cloacae* complex (ECC) is among the most common pathogens associated with nosocomial infections, causing a wide range of infectious diseases, including pneumonia, urinary tract infections, bloodstream infections, and meningitis (1). Nosocomial outbreaks involving ECC have been reported in multiple countries, including Germany (2), France (3), China (4), and Japan (5). Most strains of ECC have an intrinsic resistance to ampicillin, amoxicillin, first- and second-generation cephalosporins, and cefoxitin due to the expression of a constitutive AmpC β-lactamase. Moreover, by acquiring the mobile genetic elements, the carbapenem-resistant ECC strains were widely reported across the world (6). The mortality rate of ECC infections ranges from 22% to 40% (7, 8), with bloodstream infections posing a particularly significant threat to patient survival (9, 10).

Previous studies have shown that the ECC mainly consists of seven species: *Enterobacter cloacae, Enterobacter asburiae, Enterobacter hormaechei, Enterobacter kobei, Enterobacter ludwigii, Enterobacter mori,* and *Enterobacter nimipressuralis*. In addition, novel species, such as *Enterobacter chengduensis* (11) and *Enterobacter sichuanensis* (12), have also been classified as part of the ECC. Accurate species and subspecies identification of bacterial isolates forms the basis for understanding their epidemiology, pathogenesis, and microbiological characteristics. However, precise identification of ECC species remains challenging in clinical practice due to their highly similar genetic and biochemical characteristics. The fast development of next-generation sequencing enables further study of ECC taxonomy. Based on *hsp60* sequencing, ECC can be categorized into 12 clusters (I–XII) (13), while average nucleotide identity (ANI) analysis has further classified them into 22 clades (A–V) (14).

In this study, we retrospectively collected and identified 108 ECC strains isolated from patients with bloodstream infections over a five-year period (2015–2020) in Beijing, China. The whole-genome sequencing (WGS) was conducted to characterize the molecular features, population structure, and distribution of antimicrobial resistance genes among these isolates. Additionally, phylogenetic analysis and genomic comparison were performed to explore the genetic diversity within the ECC strains. The findings of this study provide insights into the epidemiology and resistance mechanisms of ECC, further investigating the correspondence between species, clades, and clusters, contributing to improved clinical management and infection control strategies.

## MATERIALS AND METHODS

### Clinical strain and data collection

A total of 108 ECC strains were obtained from different departments at Peking Union Medical College Hospital in the period of 2015–2020. All isolates were obtained from blood cultures and primarily identified as ECC in clinical practice using MALDI-TOF, Vitek-2, or API20. The time to positivity (TTP) of blood culture was recorded by the automated blood culture system (BD BACTEC FX) and defined as the interval between the time the blood culture bottle was loaded into the instrument and the time when a positive signal was reported by the system. Other clinical and demographic data were retrieved from the Hospital Information System of Peking Union Medical College Hospital, including patient age, sex, comorbidities, and department.

### Antimicrobial susceptibility testing

All isolates were exposed to 16 antimicrobials, including ceftriaxone, cefepime, ceftazidime, imipenem, meropenem, ertapenem, levofloxacin, colistin, polymyxin B, cefoxitin, tigecycline, eravacycline, amikacin, ceftazidime-avibactam, piperacillin-tazobactam, and sulfamethoxazole, by the broth dilution method according to the Clinical and Laboratory Standards Institute (CLSI) criteria. *Escherichia coli* ATCC 25922 was used as the minimum inhibitory concentration (MIC) reference strain for quality control, and the results were interpreted according to the CLSI breakpoint.

## Whole-genome sequencing and sequence analysis

Genomic DNA was extracted from each isolate using an AxyPrep bacterial genomic DNA miniprep kit (Axygen Scientific, Union City, CA, USA). DNA libraries with an average insert size of 400 bp were prepared using the NEBNext Ultra II DNA library preparation kit. Paired-end sequencing (2 × 150 bp) was performed on the Illumina HiSeq 2500 platform at Shanghai Sunny Biotechnology Co., Ltd. (Shanghai, China). Raw sequencing data were subjected to stringent quality control to ensure accuracy and reliability. The raw reads were initially evaluated using FastQC (v0.11.9), and quality filtering was performed with fastp (v0.20.0).

*De novo* genome assembly was carried out using SPAdes (v3.13.1) (15). Functional annotation of coding sequences was performed using Prokka (v1.13) (16). Antimicrobial resistance genes and virulence factors were identified using the ResFinder database (17). ANI values were calculated using FastANI (v1.33) (18), and a heatmap was generated to visualize ANI clustering. The type strains used for the ANI analysis are listed in Table S1 in the Supplemental File. The *hsp60* sequences of each isolate were extracted by aligning genome assemblies against *hsp60* reference sequences from NCBI. Multiple sequence alignment was performed using MAFFT (v7.450) (19), and a phylogenetic tree was constructed using MEGA11 (v11.0.13) (20) with the neighbor-joining method. Multi-locus sequence typing (MLST) was performed using the MLST database (http://pubmlst.org/ecloacae), and sequence types (STs) were assigned based on the allelic profiles of housekeeping genes. All whole-genome sequencing data from this study have been deposited in the GenBank under BioProject accession no. PRJNA1226973.

## RESULTS

### Clinical and demographic characteristics of ECC isolates

The sources of the 108 non-duplicated ECC strains include peripheral blood (*n* = 99) and catheter (*n* = 9). All isolates were identified as ECC by MALDI-TOF MS or Vitek-2 Compact during routine clinical procedures. Strains isolated from the same patient more than 7 days apart were considered distinct. The time to blood culture positivity ranged from 2 to 110 hours, with the majority (68.5%, 74/108) becoming positive within 14 hours. The average TTP was 17.6 hours.

The 108 isolates were collected from 106 patients, with two patients each providing two isolates collected at different time points. Among these patients, 55.7% (59/106) were male, and 44.3% (47/106) were female, with a mean age of 50 years. Among the 106 patients, 15.1% (16/106) were children or teenagers (0–18 years), 16.0% (17/106) were youth (19–45 years), 31.1% (33/106) were middle-aged adults (46–65 years), and 37.7% (40/106) were elderly (>65 years). Notably, middle-aged and elderly individuals (≥46 years) accounted for most cases, comprising 68.9% (73/106) of the study population.

The strains were predominantly isolated from the emergency department (20.4%, 22/108), intensive care unit (ICU) (16.7%, 18/108), pediatric department (13.9%, 15/108), and hematology department (13.9%, 15/108). The remaining strains were distributed across 16 other departments, each contributing less than 10% of the total isolates. The detailed demographic characteristics are summarized in Table S2.

### Identification of ECC isolates

WGS was performed in all the isolates, and the general indicators, including the total number of clean reads, GC content, and read quality (Q30), were indicated in Table S3. The average Q30 score across all samples exceeded 85%, indicating high base call accuracy. Additionally, the GC content of the clean reads was consistent with the expected genome composition of ECC.

The ANI results of 108 isolates revealed that the predominant species was *E. hormaechei* (*E. hormaechei* subsp. *steigerwaltii*, *n* = 28; *E. hormaechei* subsp. *oharae*, *n* = 4; *E. hormaechei*, *n* = 2), consisting of 34 strains in total (31.5%), followed by *E. xiangfangensis* (15.7%, 17/108), *E. cloacae* subsp. *cloacae* (11.1%, 12/108), *E. bugandensis*

(10.2%, 11/108), *E. hoffmannii* (10.2%, 11/108), *E. kobei* (6.5%, 7/108), *E. roggenkampii* (5.6%, 6/108), *E. ludwigii* (3.7%, 4/108), and *E. mori* (1.8%, 2/108). *E. chengduensis*, *E. asburiae*, *E. nematophilus,* and *E. cancerogenus* were identified only in one strain (0.92%) each. All isolates were grouped into 13 clades (A to G, I, J, M, Q, R, U), while two strains, E024 and E226, identified as *E. chengduensis* and *E. nematophilus,* respectively, did not belong to any clades (Table S2). Based on *hsp60* sequence analysis, the 108 isolates were classified into eleven distinct clusters (I–XI) (Fig. 1). Cluster VIII contained the largest number of isolates (28/108, 25.9%), indicating its potential role as the dominant lineage within the ECC population. This was followed by cluster VI (21/108, 19.4%), cluster III (12/108, 11.1%), and cluster XI (12/108, 11.1%). Clusters II and IV each included seven isolates (6.5%), while clusters I and V comprised four isolates each (3.7%). Cluster VII had the fewest isolates (2/108, 1.9%) (Table 1).

Among all the genotyped ECC isolates, clade-cluster mapping revealed five highly conserved lineage pairs that together accounted for more than 75% of the collection. Clade B was exclusively linked to cluster VIII, with all 28 clade B isolates (28/108, 25.9%) falling into this single cluster, indicating the most stringent clade-cluster correspondence. The next largest block was clade A in cluster VI (17/108, 15.7%). Three additional clades each exhibited a one-to-one relationship with a dominant cluster: clade G to cluster XI (12/108, 11.1%), clade D to cluster III (11/108, 10.2%), and clade R to cluster IX (11/108, 10.2%). Together, these five pairs (B-VIII, A-VI, G-XI, D-III, R-IX) comprised the majority (79/108, 73.1%) of the data set. The remaining lineages were smaller but still cluster-restricted: clade Q to cluster II (seven isolates), clade M to cluster IV (six isolates), and single-cluster micro-lineages, such as clades F, I, J, and U (≤4 isolates each, confined to clusters I or V). Cluster I was composed of three clades (clades F, G, and U), encompassing three different species (Fig. 2).

## MLST analysis

MLST analysis shows that 108 ECC strains were identified as 90 STs, including 29 novel STs (ST3001, ST3188-3199, ST3201-3215) (Table 1). Among all strains, ST527 was the most frequently detected sequence type ($n = 4$), followed by ST45, ST167, and ST346 (each with $n = 3$). Additionally, 76 unique STs were identified, each represented by a single strain. These novel ST types were found in 30 strains, with the highest representation in *E. bugandensis*, *E. cloacae* subsp. *cloacae*, and *E. hormaechei* subsp. *steigerwaltii* ($n = 5$, respectively) (Table S2).

## Antimicrobial susceptibility

The antimicrobial susceptibility profile is shown in Table 2. Among the 108 ECC isolates, high susceptibility was observed in ceftazidime-avibactam (MIC ≤ 8/4 mg/L, 100.00%), amikacin (MIC ≤ 16 mg/L, 99.1%), meropenem (MIC ≤ 2 mg/L, 95.4%), ertapenem (MIC ≤ 0.5 mg/L, 89.91%), levofloxacin (MIC ≤ 2 mg/L, 86.2%), tigecycline (MIC ≤ 1 mg/L, 87.16%), cefepime (MIC ≤ 2 mg/L, 76.15%), and piperacillin-tazobactam (MIC ≤ 16/4 mg/L, 77.1%). In contrast, lower susceptibility rates were noted for ceftazidime (MIC ≤ 4 mg/L, 69.72%), imipenem (MIC ≤ 1 mg/L, 58.72%), ceftriaxone (MIC ≤ 1 mg/L, 64.22%), and sulfamethoxazole (MIC ≤ 2/38 mg/L, 36.70%). Resistance to colistin and polymyxin B (MIC ≥ 4 mg/L) was observed in 32.1% of isolates, and the two agents shared the same resistant isolates.

Among the 20 isolates identified as carbapenem-resistant *Enterobacter cloacae* complex (CRECC), *E. roggenkampii* was the most prevalent species ($n = 5$), followed by *E. bugandensis* ($n = 3$), *E. hormaechei* subsp. *steigerwaltii* ($n = 3$), and *E. xiangfangensis* ($n = 3$), and *E. kobei* ($n = 2$). Additionally, one single CRECC isolate of *E. mori*, *E. chengduensis*, *E. cancerogenus*, and *E. cloacae* subsp. *cloacae* was also detected.

## Distribution of resistance genes among ECC isolates

A total of 404 resistance genes, representing 77 distinct types associated with resistance to 12 categories of antimicrobial agents, were identified in the genomes. These

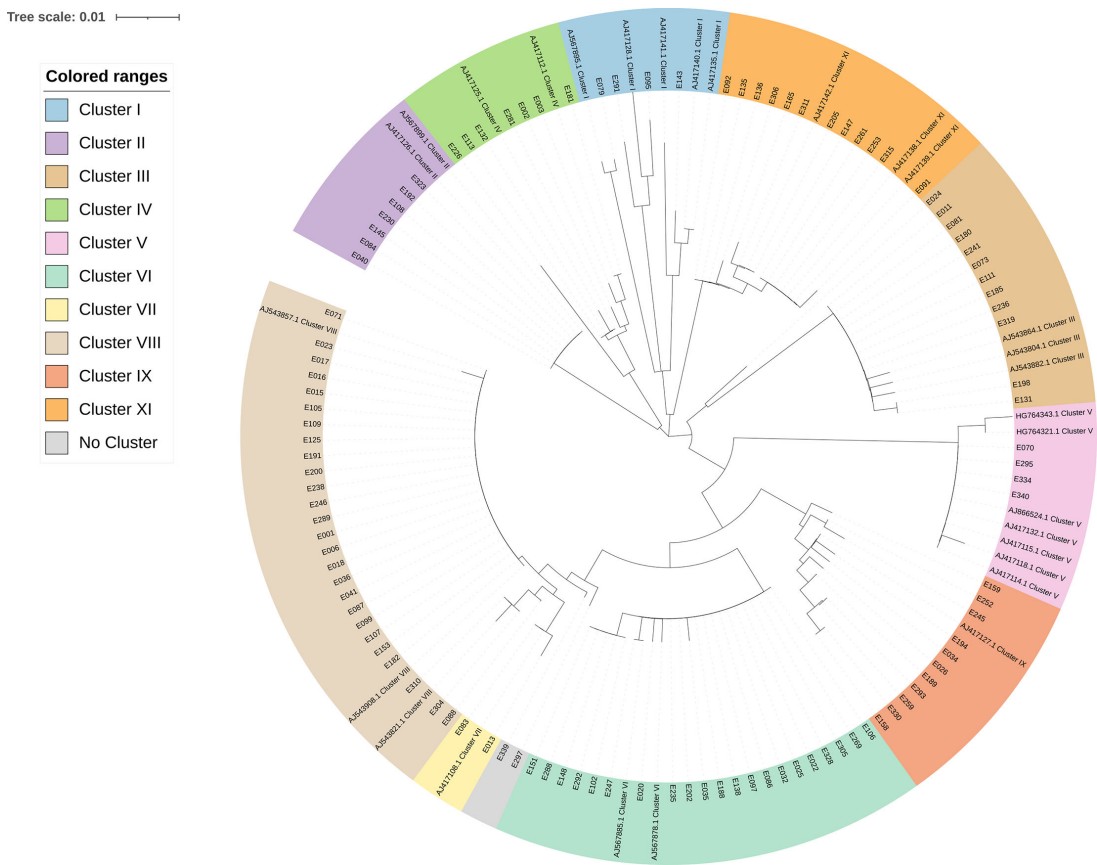

**FIG 1** Phylogenetic tree of 108 ECC isolates based on *hsp60*. The tree was constructed using the maximum likelihood method. Strain IDs are shown on the branches, with their corresponding *hsp60*-defined clusters indicated by colored sectors. Reference strains representing known *hsp60*-defined clusters are included and labeled with GenBank accession numbers and cluster names. Bootstrap values (≥50%) are indicated at the nodes.

included genes conferring resistance to aminoglycosides [e.g., *aac(3)-IIa*, *aac(3)-IId*, *aac(3)-IV*, *aac(6')-aph(2´)*, *aac(6')-Ib3*, *aac(6')-IIc*], beta-lactams (e.g., *bla*~CTX-M-15~, *bla*~NDM-1~, *bla*~SHV-12~, *bla*~TEM-1A~), fluoroquinolones (e.g., *qacE*, *qnrA1*, *qnrB1*, *qnrB4*), fosfomycin (e.g., *formA*, *fosA*, *fosA2*, *fosA3*), macrolides [e.g., *ere(A)*, *mph(A)*], phenicols (e.g., *catA2*, *catB3*, *floR*), polymyxins (*mcr-10*), quaternary ammonium compounds (*qacE*), rifampicin (*ARR-3*), sulfonamides (*sul1*, *sul2*), tetracyclines [*tet(A)*, *tet(B)*, *tet(C)*, *tet(D)*] and trimethoprim (*dfrA1*, *dfrA12*, *dfrA14*, *dfrA15*). Among these, beta-lactam resistance genes were the most prevalent, accounting for 31.4% (127/404), followed by fosfomycin resistance genes at 25.5% (103/404) and aminoglycoside resistance genes at 12.9% (52/404).

The *bla*~ACT~ genes were the most prevalent among all detected beta-lactam resistance genes, accounting for 59.8% (76/127). As shown in the heatmap, the distribution of intrinsic AmpC genes varied across different species. For instance, *bla*~ACT-16~ was found in nearly all *E. xiangfangensis* strains (94.1%, 16/17) but was absent in other species. Similarly, *bla*~ACT-12~ was detected exclusively in *E. ludwigii* (100%, 4/4), while *bla*~ACT-15~ appeared only in *E. hormaechei* subsp. *steigerwaltii* (57.1%, 16/28). The *bla*~ACT-2~ gene was identified in a single *E. asburiae* strain. Additionally, all *E. kobei* strains harbored *bla*~ACT-9~ (100%, 7/7), and *bla*~ACT-5~ was exclusively detected in *E. hoffmannii* (36.4%, 4/11). The *bla*~ACT-7~ gene exhibited a more dispersed distribution, being found in 12 strains of *E. hormaechei* subsp. *steigerwaltii* (12/28), four strains of *E. hormaechei* subsp. *oharae* (4/4), two strains of *E. hormaechei* (2/2), and one strain of *E. xiangfangensis* (1/17) (Fig. 3).

Horizontally transferable resistance genes pose a significant threat to clinical practice due to their potential to spread between bacterial species. Previous studies have demonstrated that genes, such as *bla*~CTX-M~, *bla*~TEM~, *bla*~SHV~, *bla*~NDM~, *bla*~DHA-1~, and *bla*~SFO-1~,

**TABLE 1** Summary of species, phylogenetic clades, genomic clusters, and MLST of ECC strains collected in this study[a]

| Identification | No. of strains | Clade | Cluster | MLST |
|---|---|---|---|---|
| *E. xiangfangensis* | 17 | A | VI | 1131 418 66 171 148 527 264 120 127 728 **3191** 557 |
| *E. hormaechei* subsp. *steigerwaltii* | 28 | B | VIII | 190 177 346 421 110 113 254 **3215 3189 3199** 45 664 1116 **3209** 1015 50 |
| *E. hormaechei* subsp. *oharae* | 4 | C | VI | 354 68 **3206** 108 |
| *E. hoffmanniic* | 11 | D | III | 145 135 278 233 78 **3204 3205** 518 316 102 |
| *E. hormaechei* | 2 | E | VII | 696 **3197** |
| *E. mori* | 2 | F | I | **3198 3213** |
| *E. cloacae* subsp. *cloacae* | 12 | G | XI | 524 167 **3202 3190** 1 **3211 3212** 84 **3193** 462 |
| *E. ludwigii* | 4 | I | V | **3195 3192** 12 1803 |
| *E. asburiae* | 1 | J | I | 162 |
| *E. roggenkampii* | 6 | M | IV | 703 803 501 1642 131 422 |
| *E. kobei* | 7 | Q | II | **3194 3188** 365 3201 125 691 |
| *E. bugandensis* | 11 | R | IX | 38 386 35 2301 **3207 3208 3210 3001** 1085 659 **3214** |
| *E. cancerogenus* | 1 | U | I | **3196** |
| *E. chengduensis* | 1 | –[b] | IV | 414 |
| *E. nematophilus* | 1 | – | III | 2769 |

[a]Boldface indicates novel STs identified in this study.
[b]"–" indicates that the isolates do not belong to any Clade or Cluster.

can be mobilized via plasmids or other mobile genetic elements. In total, 28 horizontally transferable resistance genes were identified across 15 strains. The most prevalent of these was $bla_{TEM-1B}$ (6/28), followed by $bla_{CTX-M-3}$ (4/28), $bla_{OXA-1}$ (4/28), and $bla_{CTX-M-15}$ (3/28). Notably, 41.2% of *E. xiangfangensis* strains (7/17) carried at least one horizontally transferable gene. Among these, strains E086 and E235 harbored four such genes simultaneously. Specifically, E086 contained $bla_{DHA-1}$, $bla_{SFO-1}$, $bla_{SHV-12}$, and $bla_{TEM-1B}$, while E235 carried $bla_{CTX-M-15}$, $bla_{NDM-1}$, $bla_{OXA-1}$, and $bla_{SHV-12}$. Unlike intrinsic resistance genes, these horizontally transferable genes showed no clear species-specific distribution patterns.

Regarding carbapenem resistance genes, $bla_{NDM-1}$ and $bla_{NDM-5}$ were detected in only one strain (E235) and two strains (E041 and E071), respectively. Strains E041 and E071 were identified as *E. hormaechei* subsp. *steigerwaltii*, while E235 belonged to *E. xiangfangensis*. All three strains exhibited high-level resistance to carbapenems, with MICs ≥32 mg/L for ertapenem, imipenem, and meropenem, as well as resistance to fourth-generation cephalosporins. In total, 20 isolates in this study exhibited carbapenem-resistant phenotypes. Notably, the majority of them (17/20, 85.0%) did not carry any known carbapenemase genes. Among these 17 non-carbapenemase-producing strains, 82.4% (14/17) were resistant to only one carbapenem agent (either ertapenem or imipenem) and exhibited relatively low MICs (2 or 4 mg/L). Strains E113 and E259, which were resistant to all tested carbapenems and showed relatively high MICs (≥16 mg/L for ertapenem), also lacked detectable carbapenemase genes. However, E113 OmpC contained a total of 13 amino acid substitutions and one deletion compared to *E. roggenkampii* DSM 16690. Three alterations were located in the N-terminal signal peptide (M1V, K2V, and deletion of L3). In the β-barrel region, eight scattered substitutions were detected, including R152P, D198T, and I225A. Importantly, the most pronounced changes were identified within the constriction loop L3 (residues 267–273), where the reference motif SLTYD was replaced by SVTYN, involving four amino acid substitutions. While in E259, it encoded a truncated OmpA protein of only 180 amino acids compared to the *E. bugandensis* EB-247 reference sequence of 351 amino acids. A premature stop codon resulted in the complete loss of the C-terminal region. This truncated variant may impair outer membrane stability, then directly affect antimicrobial permeability.

A total of 15 tetracycline destructase genes were identified in 14 isolates, including *tet*(A), *tet*(B), *tet*(C), and *tet*(D), with *tet*(A) being the most prevalent (7/15, 46.7%). Among these *tet*-positive strains, only seven exhibited phenotypic resistance to tigecycline.

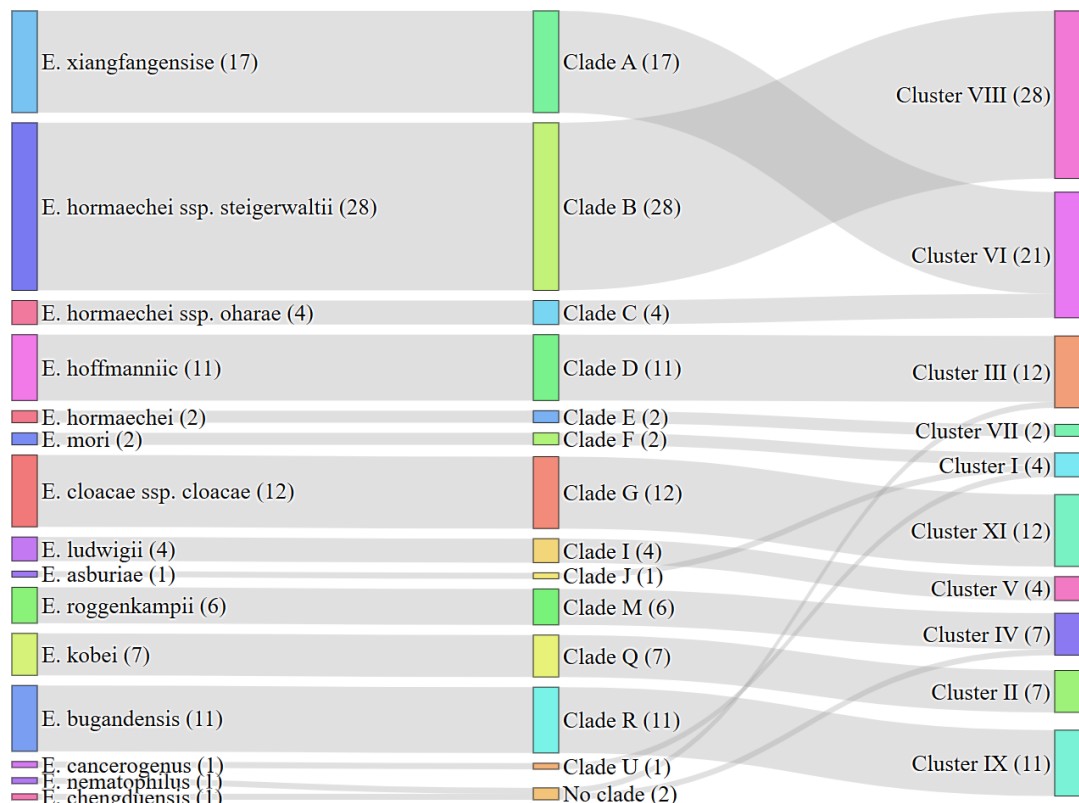

FIG 2 Relationships among species, phylogenetic clades, and genomic clusters of ECC isolates. The Sankey diagram illustrates the hierarchical relationship among 108 ECC isolates at three levels: species (left), clades (middle), and clusters (right). Numbers in parentheses indicate the number of isolates within each category.

While there were 14 strains resistant to tigecycline in total, seven strains of them harbored no destructase genes.

Polymyxins have been reintroduced in recent years as a last-resort treatment for CRECC. In this study, 35 isolates were resistant to both polymyxin B and colistin (polymyxin E). Among all polymyxin-resistant strains, only one (E306) was detected to harbor a known resistance gene, *mcr-10*, and exhibited high-level resistance to both polymyxin B and colistin (MIC > 64 mg/L).

## Genome analysis of strains from the same patient

There are a total of four strains that were isolated from two patients (labeled as P1 and P2), each patient with two strains, respectively. The clinical and microbiological information of these strains is shown in Table 3.

Each pair of strains shows different kinds of single-nucleotide polymorphisms (SNPs) in the genome and antimicrobial susceptibility patterns. For the isolates obtained from patient P1, the blood sample yielding strain E151 was collected 11 days after that of E148. During this period, the patient's condition deteriorated, leading to a transfer from the Department of General Medicine to the ICU. Notably, the two strains exhibited marked differences in antimicrobial susceptibility profiles, and further genomic analysis was conducted. E148, the earlier collected isolate, was used as the reference genome to detect within-host genomic variations, allowing accurate identification of SNPs acquired in E151. As shown in Table 3, compared to E148, E151 displayed markedly higher resistance to third- and fourth-generation cephalosporins. Additionally, E151 harbored a greater number of resistance genes that were absent in E148, including *aadA3, ant(2″)-Ia, bla*$_{CTX-M-3}$*, bla*$_{CTX-M-9}$*, and sul1*. Using the E148 assembly as a reference genome, a total of 152 missense SNPs and one missense indel mutation were identified in E151. Notably, 39

**TABLE 2** Antimicrobial susceptibility profile of ECC isolates

| Antimicrobial | MIC50 (mg/L) | MIC90 (mg/L) | % Susceptible | % Intermediate | % Resistant |
|---|---|---|---|---|---|
| Ceftriaxone | 0.5 | >128 | 69 (64.22%) | 1 (0.92%) | 38 (34.86%) |
| Cefepime | <0.064 | 128 | 82 (76.15%) | 10 (9.17%) | 16 (14.68%) |
| Ceftazidime | 0.25 | >128 | 75 (69.72%) | 2 (1.83%) | 31 (28.44%) |
| Imipenem | 1 | 4 | 64 (58.72%) | 27 (25.69%) | 17 (15.60%) |
| Meropenem | <0.064 | 64 | 103 (95.41%) | 1 (0.92%) | 4 (3.67%) |
| Ertapenem | 0.25 | 64 | 97 (89.91%) | 3 (2.75%) | 8 (7.34%) |
| Levofloxacin | 0.064 | 4 | 93 (86.24%) | 4 (3.67%) | 11 (10.09%) |
| Colistin | <0.25 | >64 | $-^{a}$ | 74 (67.89%) | 35 (32.11%) |
| Polymyxin | <0.25 | >64 | – | 74 (67.89%) | 35 (32.11%) |
| Tigecycline | 0.5 | 2 | 94 (87.16%) | 0 | 14 (12.84%) |
| Eravacycline | 0.5 | 2 | – | – | – |
| Amikacin | 1 | 4 | 107 (99.08%) | 1 (0.92%) | 0 |
| Ceftazidime-avibactam | <0.064/4 | <0.064/4 | 108 (100.00%) | 0 | 0 |
| Piperacillin-tazobactam | <8/4 | 16/4 | 83 (77.06%) | 10 (9.17%) | 15 (13.76) |
| Sulfamethoxazole | >4/76 | >4/76 | 40 (36.70%) | 0 | 68 (63.30%) |

[a]"–" indicates no breakpoint available / not interpretable for ECC.

of these 152 SNPs were located in the *pcoE* and *pcoS* genes, which are components of the copper tolerance system.

The two strains isolated from P2, E292 and E295, were different species; also, they were identified as different clades and MLST. The isolation time interval between these two strains was 35 days, which means the patient P2 was infected with two different strains during his stay.

## DISCUSSION

ECC is a key member of the ESKAPE pathogens (*Enterococcus faecium*, *Staphylococcus aureus*, *Klebsiella pneumoniae*, *Acinetobacter baumannii*, *Pseudomonas aeruginosa*, and *Enterobacter* spp., ESKAPE), ranking among the most prevalent species causing nosocomial infections. However, accurate species and subspecies identification within the ECC remains challenging in clinical practice. A better understanding of their molecular epidemiology is fundamental for effective treatment and infection control. Previous studies have shown that ECC can be classified into distinct clusters based on *hsp60* sequence analysis (13). However, with the rapid development of high-resolution typing techniques, such as WGS and ANI analysis, more accurate and detailed taxonomic classifications have been achieved (2). In this study, *E. hormaechei* (31.5%, 34/108) and *E. xiangfangensis* (15.7%, 17/108) were the predominant species, consistent with previous clinical reports (21). Notably, a previous study in southwestern China found *E. xiangfangensis* to be the most common species among bloodstream infection isolates (21/48, 43.8%) (22), which differs slightly from our findings, potentially reflecting regional population differences.

A total of 90 STs were identified, and 29 out of them were novel STs, with ST527 being the most prevalent type. It has been proven that ECC has high genetic diversity, and the same novel ST can be found in different countries, which may be associated with resistance spread (23). These newly defined types in this study suggest ongoing genomic evolution and the possibility of underreported lineages within the ECC population. Previous studies have demonstrated that certain clones, such as ST171 and ST78, are more likely to carry plasmids encoding carbapenemases (24). In our study, only four isolates belonged to these clones (ST171, *n* = 2; ST78, *n* = 2), and among them, only one strain (E235) was resistant to carbapenems. Notably, the 20 CRECC isolates identified in this study were distributed across 19 different STs, suggesting no dominant ST among CRECC strains in this collection.

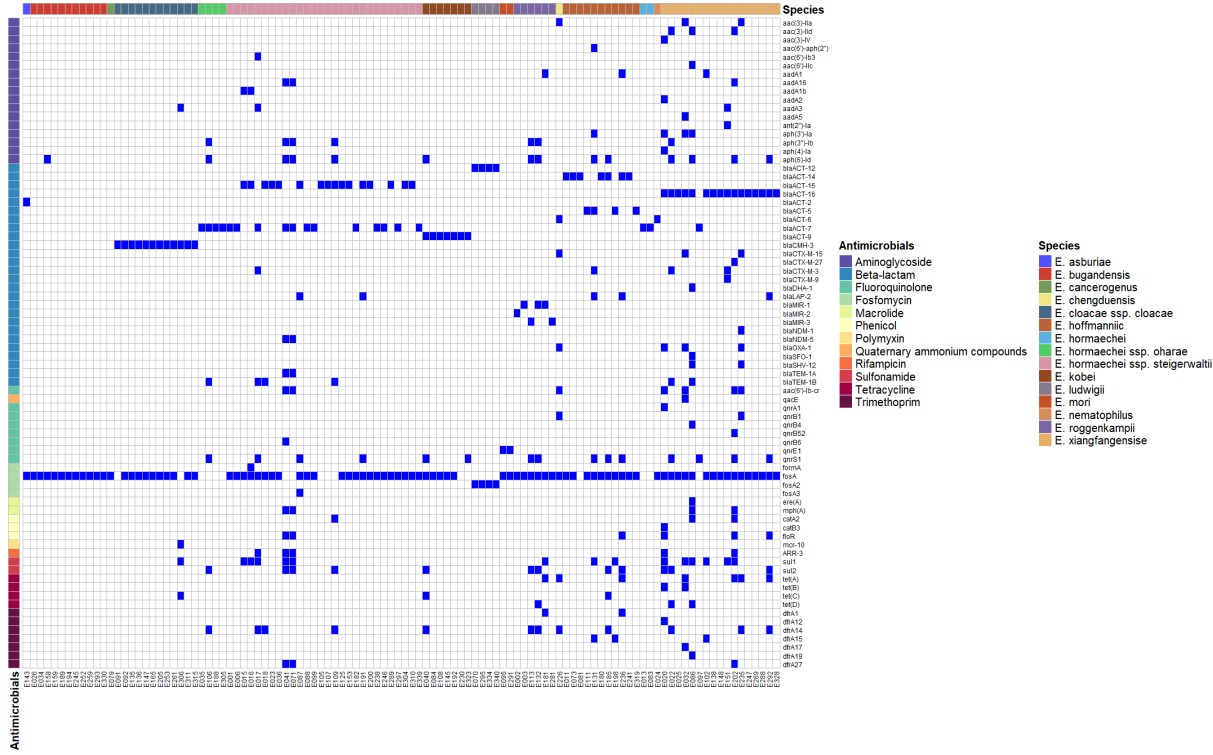

**FIG 3** Distributions of antimicrobial genes in 108 ECC isolates. This heatmap illustrates the presence or absence of acquired antimicrobial resistance genes across 108 ECC isolates. Each column represents one isolate, and each row corresponds to a specific resistance gene. Blue squares indicate the presence of the gene, while white squares indicate its absence.

Hoffmann and Roggenkamp (13) defined 12 genetic clusters (I to XII) based most exhaustively on *hsp60* sequencing. Three of the clusters (cluster III, cluster VI, cluster VIII) accounted for most of the strains studied. The cluster typing results in our study generally aligned with WGS-based species identification, consistent with the classification proposed by Hoffmann (25, 26). In our analysis, several distinct clades were grouped into the same *hsp60*-based cluster. For instance, clade A (*E. xiangfangensis*) and clade C (*E. hormaechei* subsp. *oharae*) were both classified into cluster VI, while clade D (*E. hoffmannii*) and *E. nematophilus* were both assigned to cluster III. This highlights the limited resolution of *hsp60*-based typing in distinguishing closely related genomic lineages. In contrast, whole-genome-based clade typing provides better phylogenetic resolution. Meanwhile, two recently described species, *E. chengduensis* and *E. nematophilus*, were not assigned to any defined clade, suggesting that the current clade classification needs to be further improved. These findings underline the need for continuous refinement of clade-based frameworks as more genomic data becomes available.

Most ECC isolates are known to produce chromosomally encoded AmpC β-lactamases. In our study, *bla*$_{ACT}$ genes were the most prevalent β-lactamase genes, detected in 59.8% of isolates, and demonstrated clear species specificity. The *bla*$_{ACT-16}$, *bla*$_{ACT-12}$, *bla*$_{ACT-2}$, *bla*$_{ACT-9}$, and *bla*$_{ACT-5}$ were exclusively identified in *E. xiangfangensis, E. ludwigii, E. asburiae, E. kobei,* and *E. hoffmannii*, respectively. This finding is highly consistent with a previous study (21), yet provides more detailed resolution regarding gene-subtype associations across species.

In our study, three strains were identified as carrying carbapenemase genes, including two harboring *bla*$_{NDM-5}$ (*E. hormaechei* subsp. *steigerwaltii*) and one harboring *bla*$_{NDM-1}$ (*E. xiangfangensis*). In contrast to our findings, other epidemiological studies of ECC in China reported that bla$_{NDM}$ was the most prevalent carbapenemase gene, with *bla*$_{NDM-5}$ reported exclusively in *E. xiangfangensis* (21, 27, 28). Moreover, some studies have indicated that *bla*$_{NDM-1}$ is the most prevalent variant among carbapenem-resistant

**TABLE 3** Serial ECC isolates from patients and the antimicrobial resistance profiles[a]

| Patient | Gender | Age | Diagnosis | Department | No. | Date | Strain | Clade | MLST | Resistance gene | CRO | FEP | SMX | CZA | IPM | MEM | ETP | TZP | LVX | POL | CST | FOX | TGC | ERV | AMK | SMX |
|---|---|---|---|---|---|---|---|---|---|---|---|---|---|---|---|---|---|---|---|---|---|---|---|---|---|---|
| P1 | M | 33 | Systemic lupus erythematosus | General internal medicine | E148 | 2017/7/16 | E. xiangfangensise | A | 527 | $bla_{ACT-16}$, fosA | 0.25 | 0.125 | 0.5 | <0.064/4 | 1 | <0.064 | 0.25 | <8/4 | 0.125 | <0.25 | <0.25 | >64 | 2 | 2 | 1 | >4/76 |
| | | | | ICU | E151 | 2017/7/27 | E. xiangfangensise | A | 527 | aadA3, ant(2'')-Ia, $bla_{ACT-16}$, $bla_{CTX-M-3}$, $bla_{CTX-M-9}$, fosA, sul1 | >128 | 32 | 2 | <0.064/4 | 1 | <0.064 | 0.25 | <8/4 | 0.125 | <0.25 | <0.25 | >64 | 2 | 2 | 2 | >4/76 |
| P2 | M | 37 | Lymphoma | Hematology | E292 | 2020/4/15 | E. xiangfangensise | A | 3191 | aph(6)-Id, $bla_{ACT-16}$, $bla_{LAP-2}$, dfrA14, floR, fosA, qnrS1, sul2, tet(A) | 0.125 | <0.064 | 0.25 | <0.064/4 | 1 | <0.064 | 0.25 | <8/4 | 0.5 | 0.5 | <0.25 | >64 | 1 | 0.5 | 2 | >4/76 |
| | | | | | E295 | 2020/5/20 | E. ludwigii | I | 3192 | $bla_{ACT-12}$, fosA2 | 0.125 | 0.125 | 0.25 | <0.064/4 | 2 | <0.064 | 0.25 | <8/4 | 0.032 | <0.25 | <0.25 | >64 | 0.5 | 0.25 | 2 | 4/76 |

[a]CRO, ceftriaxone; FEP, cefepime; CAZ, ceftazidime; CZA, ceftazidime-avibactam; IPM, imipenem; MEM, meropenem; ETP, ertapenem; TZP, piperacillin-tazobactam; LVX, levofloxacin; PMX, polymyxin; CST, colistin; FOX, cefoxitin; TGC, tigecycline; ERV, eravacycline; AMK, amikacin; SMX, sulfamethoxazole.

isolates in China (29, 30), which is also different from our findings. These differences may be attributed to geographical variation or the limited number of isolates analyzed in our study. Notably, most of the carbapenem-resistant isolates (17/20, 85.0%) have no carbapenemase genes. Among them, 82.4% isolates (14/17) were resistant to only one kind of carbapenem (ertapenem or imipenem). *E. roggenkampii* was the most frequently detected species among CRECC isolates (5/20, 25.0%). Of the six *E. roggenkampii* isolates identified, five (83.3%) exhibited a carbapenem-resistant phenotype. Previous genomic analysis showed that the production of an AmpC-type cephalosporinase of the MIR family is highly prevalent in *E. roggenkampii* (31), possibly resulting in the carbapenem-resistant phenotype combined with truncation of the gene for an OmpC-type porin (32). In addition, more than 80% of *E. roggenkampii* isolates have been reported to be resistant to colistin in previous studies (33). Consistent with this, four out of six *E. roggenkampii* strains in our study exhibited resistance to both colistin and polymyxin B. However, the overall rate of carbapenem resistance in *E. roggenkampii* remains poorly defined in the literature. The high proportion of isolates simultaneously resistant to both carbapenems and colistin observed here underscores the clinical importance of *E. roggenkampii* and calls for increased surveillance and infection control strategies.

Discrepancies were observed between the presence of *tet* genes and phenotypic resistance to tigecycline. Several isolates harboring *tet* genes remained susceptible to tigecycline, while others exhibited phenotypic resistance in the absence of detectable *tet* determinants. These inconsistencies may be attributed to several factors. First, not all *tet* genes confer resistance to tigecycline; for instance, *tet(A)* and *tet(B)* may mediate low-level resistance or be inactive due to truncations or regulatory mutations (34). Second, tigecycline resistance is often multifactorial and may involve overexpression of efflux pumps, such as AcrAB-TolC (35), or mutations in global regulators like *ramR*, *marR*, or *ramA* (36). Therefore, the presence or absence of *tet* genes alone may not fully explain tigecycline susceptibility, and comprehensive analysis, including efflux mechanisms and transcriptional regulation, may be required for accurate interpretation. With regard to polymyxin resistance, among all polymyxin-resistant isolates, only one (E306) was found to harbor a known resistance gene, *mcr-10*, which is associated with plasmid-mediated lipid A modification and conferred high-level resistance (MIC > 64 mg/L) (37). The absence of detectable *mcr* genes in the remaining resistant strains suggests the involvement of chromosomal mutations, particularly in two-component regulatory systems, such as *phoP/phoQ*, *pmrA/pmrB*, or disruptions in the *mgrB* gene, which are known to alter the structure of lipid A and reduce polymyxin binding affinity (38).

Isolates collected from the same patient at different time points revealed evidence of within-host evolution during the treatment. In this study, we collected two pairs of isolates from the same patients. Changes in antimicrobial resistance phenotypes were observed between the sequential isolates, E148 and E151, suggesting that antimicrobial pressure may have driven genetic adaptation. By comparing the genomes of strains E148 and E151 isolated from patient P1, a total of 152 SNPs were identified. Although the short time interval between the isolation of E148 and E151 suggests potential antimicrobial-driven selection, the acquisition of five additional resistance genes and over one hundred SNPs may also result from horizontal gene transfer or homologous recombination events. Alternatively, these isolates could represent co-infection by two genetically distinct *E. xiangfangensis* lineages rather than microevolution within a single strain. A significant proportion of these mutations were located within the *pcoE* and *pcoS* genes, which are components of the *pco* operon, which is known for its role in bacterial resistance to copper and silver ions (39). Several studies have demonstrated that copper-cephalosporin complexes exhibit enhanced antimicrobial activity compared to cephalosporins alone (40–42). In Gram-negative bacteria, copper resistance is often mediated by efflux pump systems, which are also commonly involved in decreased susceptibility to antimicrobial agents (43). Although no direct evidence currently links copper resistance mechanisms to cephalosporin resistance, these mutations may

indirectly contribute to β-lactam resistance. The specific mechanisms underlying this association remain to be fully elucidated.

In conclusion, we conducted the precise species identification and whole-genome sequencing analysis of 108 *Enterobacter* bloodstream infection isolates in China. To our knowledge, this is one of the largest genomic epidemiological studies of ECC bloodstream isolates in northern China, revealing a highly diverse population structure and discovering 29 novel sequence types. *E. hormaechei* and *E. xiangfangensis* were identified as the dominant species. In terms of the distribution of resistance genes, chromosomally encoded $bla_{ACT}$ genes showed great species specificity. Notably, *E. roggenkampii* emerged as a key species associated with carbapenem and colistin resistance, which has not been well-characterized in prior surveillance studies. Genomes of isolates from the same patients demonstrated within-host evolution, including the acquisition of additional resistance genes and functional mutations in genes, such as *pcoE/pcoS*. These findings not only advance our understanding of ECC pathogenesis and resistance evolution in China but also clarify the correspondence between species, clades, and clusters, offering a valuable genomic resource and new perspective for taxonomy and rapid diagnostic development.

## ACKNOWLEDGMENTS

This work was supported by the National Science Foundation for Young Scientists of China (82202541), the Peking Union Medical College Hospital Talent Cultivation Program Category D (UHB12396), and the Fundamental Research Funds for the Central Universities (3332022012).

## AUTHOR AFFILIATIONS

[1]Department of Clinical Laboratory, State Key Laboratory of Complex Severe and Rare Diseases, Peking Union Medical College Hospital, Chinese Academy of Medical Sciences and Peking Union Medical College, Beijing, People's Republic of China
[2]Graduate School, Peking Union Medical College, Chinese Academy of Medical Sciences, Beijing, People's Republic of China

## AUTHOR ORCIDs

Yanbing Li ⓘ http://orcid.org/0009-0003-4141-8534
Yingchun Xu ⓘ http://orcid.org/0000-0002-7126-9459
Menglan Zhou ⓘ http://orcid.org/0000-0003-1241-6153

## FUNDING

| Funder | Grant(s) | Author(s) |
| --- | --- | --- |
| National Science Fund for Distinguished Young Scholars | 82202541 | Menglan Zhou |
| Peking Union Medical College Hospital | UHB12396 | Menglan Zhou |
| Central University Basic Research Fund of China | 3332022012 | Menglan Zhou |

## ETHICS APPROVAL

This study was reviewed and approved by the Ethics Committee of Peking Union Medical College Hospital (approval no. I-22PJ396). Only identified ECC isolates collected during routine clinical diagnostics were used. Therefore, the requirement for informed consent was waived by the Ethics Committee in accordance with the "Ethical Review Measures for Biomedical Research Involving Humans" and the "Regulations of the People's Republic of China on the Administration of Human Genetic Resources." The study was conducted in accordance with the ethical principles of the Declaration of Helsinki.

## ADDITIONAL FILES

The following material is available online.

## Supplemental Material

**Supplemental material (Spectrum02865-25-S0001.xlsx).** Tables S1 to S3.

## Open Peer Review

**PEER REVIEW HISTORY (review-history.pdf).** An accounting of the reviewer comments and feedback.

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
