## [Reviewer comments · Microbiology Spectrum]

Microbiology Spectrum

Precise Species Identification and Whole-Genome Sequencing Analysis of *Enterobacter cloacae* complex Causing Bloodstream Infections in China

Yanbing Li, Ziran Wang, Ge Zhang, Wei Kang, Jin Li, Ying-Chun Xu, and Menglan Zhou

Corresponding Author(s): Ying-Chun Xu, Peking Union Medical College Hospital

Review Timeline:

Submission Date:	September 10, 2025
Editorial Decision:	October 28, 2025
Revision Received:	November 8, 2025
Accepted:	November 18, 2025

Editor: Vittal Ponraj

Reviewer(s): The reviewers have opted to remain anonymous.

Transaction Report:

DOI: <https://doi.org/10.1128/spectrum.02865-25>

Re: Spectrum02865-25 (**Precise Species Identification and Whole-Genome Sequencing Analysis of *Enterobacter cloacae* complex Causing Bloodstream Infections in China**)

Dear Prof. Ying-Chun Xu:

Thank you for the privilege of reviewing your work. Below you will find my comments, instructions from the Spectrum editorial office, and the reviewer comments.

Revision Guidelines

Sincerely,
Vittal Ponraj
Editor
Microbiology Spectrum

Reviewer #1 (Comments for the Author):

The authors describe the results of studies to investigate the phylogenetics of ECC isolates using whole genome sequencing. The study addresses an important aspect of clinical microbiology given the significance of ECC in nosocomial infections and possible correlation of species with intrinsic antimicrobial resistance. The experimental approach is appropriate and scientifically sound.

Most of the authors' conclusions are well supported by the presented data. The conclusion in Lines 371-372 that that "antibiotic

pressure may have driven genetic adaptation" in patient P1 needs further discussion. In particular, given the relatively large number of differences between E148 and E151 (152 SNPs and 5 resistance genes), the authors should include a brief discussion of a genetic mechanism for incorporation of so many changes in a strain over a few days. Also, the authors should mention the possibility of infection with two strains in addition to genetic adaptation.

Line 274: Is "MS148" a typographical error? If not, please describe MS148 as this is the only place it appears in the manuscript.

The study includes some patient information, but the authors did not mention IRB approval or informed consent.

Reviewer #2 (Comments for the Author):

Spectrum02865-25

The *Enterobacter cloacae* complex is among the ESKAPE pathogens and represents a major cause of hospital-acquired infections, with reported mortality rates ranging from 20% to 46%. This bacterial group has been implicated in outbreaks across multiple countries. Most *E. cloacae* complex strains exhibit intrinsic resistance to first- and second-generation cephalosporins, ceftazidime, ampicillin, and amoxicillin, thereby limiting available treatment options. In this study, the authors employed whole-genome sequencing to investigate the molecular epidemiology and resistance mechanisms of *E. cloacae* complex isolates recovered from bloodstream infections between 2015 and 2020. The findings are expected to contribute to improved patient management and enhanced infection control strategies. Below are my comments:

Major comments:

Line 274-278: Please clarify why E148 was chosen as the reference genome for SNP analysis. Ideally, a well-annotated reference genome from the NCBI database, preferably a susceptible strain, should be used for such analyses. Using a strain that is not well annotated may lead to inaccurate or misleading results.

Minor comments:

Line 2: Correct the spelling of "causing" (currently written as "casuing").

Line 26: Rephrase, it is inaccurate to state that MLST was performed. The MLST data were extracted from the WGS, rather than generated independently.

Line 87: Rephrase to "clinical strains and data collection."

Line 96: Maintain consistency throughout the manuscript, use either "antimicrobial" or "antibiotic", not both interchangeably.

Line 96: Replace "test" with "testing."

Line 103: Remove the unnecessary full stop.

Line 117: Include a reference for the hsp60 strain obtained from NCBI.

Lines 127-129 and 293-295: These sentences are repetitive and should be removed, as the same information is already stated in lines 77-78.

Line 132-134: Standardize time units, use either "h" or "hours" consistently throughout.

Line 135: Provide exact figures (e.g., 56.1% (5/10)).

Line 136: Clarify the discrepancy, 106 patients are mentioned; please specify what happened to the remaining 2 patients.

Line 181: Review wording, ST527 (N=4) may not justify the term "predominant" if only four isolates were detected.

Lines 187-188: Standardize terminology, use either "antibiotic" or "antimicrobial" consistently.

Lines 189 and 191: Include the MIC ranges defining high and low resistance rates.

Line 263: Rephrase the sentence beginning with "about" for clarity.

Line 274: Correct "MS148" to "E148."

Line 311: Rephrase or remove "it was" for sentence flow.

Figure S1: Add color coding to enhance clarity, and consider moving this figure into the main manuscript instead of leaving it in the supplementary section.

Spectrum02865-25

The *Enterobacter cloacae* complex is among the ESKAPE pathogens and represents a major cause of hospital-acquired infections, with reported mortality rates ranging from 20% to 46%. This bacterial group has been implicated in outbreaks across multiple countries. Most *E. cloacae* complex strains exhibit intrinsic resistance to first- and second-generation cephalosporins, ceftazidime, ampicillin, and amoxicillin, thereby limiting available treatment options. In this study, the authors employed whole-genome sequencing to investigate the molecular epidemiology and resistance mechanisms of *E. cloacae* complex isolates recovered from bloodstream infections between 2015 and 2020. The findings are expected to contribute to improved patient management and enhanced infection control strategies. Below are my comments:

Major comments:

Line 274-278: Please clarify why E148 was chosen as the reference genome for SNP analysis. Ideally, a well-annotated reference genome from the NCBI database, preferably a susceptible strain, should be used for such analyses. Using a strain that is not well annotated may lead to inaccurate or misleading results.

Minor comments:

Line 2: Correct the spelling of “causing” (currently written as “casuing”).

Line 26: Rephrase, it is inaccurate to state that MLST was performed. The MLST data were extracted from the WGS, rather than generated independently.

Line 87: Rephrase to “clinical strains and data collection.”

Line 96: Maintain consistency throughout the manuscript, use either “antimicrobial” or “antibiotic”, not both interchangeably.

Line 96: Replace “test” with “testing.”

Line 103: Remove the unnecessary full stop.

Line 117: Include a reference for the *hsp60* strain obtained from NCBI.

Lines 127–129 and 293–295: These sentences are repetitive and should be removed, as the same information is already stated in lines 77–78.

Line 132–134: Standardize time units, use either “h” or “hours” consistently throughout.

Line 135: Provide exact figures (e.g., 56.1% (5/10)).

Line 136: Clarify the discrepancy, 106 patients are mentioned; please specify what happened to the remaining 2 patients.

Line 181: Review wording, ST527 (N=4) may not justify the term “predominant” if only four isolates were detected.

Lines 187–188: Standardize terminology, use either “antibiotic” or “antimicrobial” consistently.

Lines 189 and 191: Include the MIC ranges defining high and low resistance rates.

Line 263: Rephrase the sentence beginning with “about” for clarity.

Line 274: Correct “MS148” to “E148.”

Line 311: Rephrase or remove “it was” for sentence flow.

Figure S1: Add color coding to enhance clarity, and consider moving this figure into the main manuscript instead of leaving it in the supplementary section.

Response to Reviewer #1

Thank you for the careful reading of our manuscript and for the constructive comments that have helped us improve the quality of the paper. We provide detailed responses to each point below, with the corresponding revisions indicated in the manuscript.

Comment 1:

Most of the authors' conclusions are well supported by the presented data. The conclusion in Lines 371-372 that that "antibiotic pressure may have driven genetic adaptation" in patient P1 needs further discussion. In particular, given the relatively large number of differences between E148 and E151 (152 SNPs and 5 resistance genes), the authors should include a brief discussin of a genetic mechanism for incorporation of so many changes in a strain over a few days. Also, the authors should mention the possibility of infection with two strains in addition to genetic adaptation.

Response:

Thank you for the comment. In the revised manuscript, we have expanded the discussion to address possible mechanisms underlying the genomic changes observed between isolates E148 and E151. Specifically, we now note that horizontal gene transfer or recombination events may explain the appearance of multiple additional resistance genes within a short time. Furthermore, we have mentioned the alternative possibility that patient P1 may have been infected or colonized by two distinct *E. xiangfangensis* strains rather than a single strain undergoing *in vivo* evolution. The revised text is included in the Discussion section (Lines 377-381 in the clean version).

Comment 2:

Line 274: Is "MS148" a typographical error? If not, please describe MS148 as this is the only place it appears in the manuscript.

Response:

Thank you for pointing this out and sorry for the misleading. "MS148" was a typographical mistake and has been corrected to "E148" (Lines 282 in the clean

version).

Comment 3:

The study includes some patient information, but the authors did not mention IRB approval or informed consent.

Response:

We appreciate the reviewer's attention to this point. As our study involved retrospective analysis of identified bacterial isolates obtained as part of routine diagnostic testing, the need for informed consent was waived. We have now added a statement regarding ethics approval and relevant legislation in the Materials and Methods section as follows (Lines 122-129 in the clean version).

Response to Reviewer #2

Thank you for the careful reading of our manuscript and for the constructive comments that have helped us improve the quality of the paper. We provide detailed responses to each point below, with the corresponding revisions indicated in the manuscript.

Major Comment

Comment 1:

Lines 274-278: Please clarify why E148 was chosen as the reference genome for SNP analysis. Ideally, a well-annotated reference genome from the NCBI database, preferably a susceptible strain, should be used for such analyses. Using a strain that is not well annotated may lead to inaccurate or misleading results.

Response:

Thank you for this important suggestion. The main purpose of this analysis was to explore within-host evolution between sequential isolates (E148 and E151) from patient P1. Therefore, we used the earlier isolate (E148) as the reference to identify genetic changes within the host. We agree that using a well-annotated reference genome can provide more genomic information and explore more resistance mechanisms. However, using E148 as the baseline allowed a direct comparison of specific mutations and minimized potential alignment artifacts and false-positive SNP that probably arise when mapping reads to a more distantly related reference genome. To clarify this, we have revised the text in the Results section as follows (Lines 275-278 in the clean version)

Minor Comments

Comment 2: Line 2: Correct the spelling of "causing" (currently written as "casuing").

Response: Thank you for your comment. Corrected to "causing" in the title (Line 2).

Comment 3: Line 26: Rephrase, it is inaccurate to state that MLST was performed. The MLST data were extracted from the WGS, rather than generated independently.

Response: Thank you for your comment. Revised to: Multi-locus sequence typing

(MLST) profiles were extracted from the WGS data (Line 26 in the clean version).

Comment 4: Line 87: Rephrase to "clinical strains and data collection."

Response: Thank you for your comment. Modified as suggested. The section heading now reads Clinical strain and data collection (Line 86 in the clean version).

Comment 5: Line 96: Maintain consistency throughout the manuscript, use either "antimicrobial" or "antibiotic", not both interchangeably.

Response: Thank you for your comment. We standardized the term and now consistently use "antimicrobial" throughout the manuscript.

Comment 6: Line 96: Replace "test" with "testing."

Response: Thank you for your comment. Corrected to "Antimicrobial susceptibility testing" as suggested (Line 95 in the clean version).

Comment 7: Line 103: Remove the unnecessary full stop.

Response: Thank you for your comment. The redundant full stop has been removed (Line 102 in the clean version).

Comment 8: Line 117: Include a reference for the *hsp60* strain obtained from NCBI.

Response: Thank you for the suggestion. The accession numbers of the *hsp60* reference sequences retrieved from NCBI of different clusters have been indicated directly in the newly added Figure 3. This visual reference provides clear traceability without duplicating the information in the Methods section.

Comment 9: Lines 127-129 and 293-295: These sentences are repetitive and should be removed, as the same information is already stated in lines 77-78.

Response: Thank you for your comment. The repeated sentences have been deleted to avoid redundancy in both lines.

Comment 10: Line 132-134: Standardize time units, use either "h" or "hours" consistently throughout.

Response: Thank you for your comment. All time units have been standardized to "hours".

Comment 11: Line 135: Provide exact figures (e.g., 56.1% (5/10)).

Response: Thank you for your comment. Percentages are now accompanied by numbers (Lines 140-141 in the clean version)

Comment 12: Line 136: Clarify the discrepancy, 106 patients are mentioned; please specify what happened to the remaining 2 patients.

Response: Thank you for your comment. In this study, there are a total of four strains that were isolated from two patients, resulting in 108 isolates from 106 patients. We clarified that in Lines 139-140 in the clean version.

Comment 13: Line 181: Review wording, ST527 (N=4) may not justify the term "predominant" if only four isolates were detected.

Response: Thank you for your comment. We revised wording to "ST527 was the most frequently detected sequence type (n=4)" (Lines 187-188 in the clean version).

Comment 14: Lines 187-188: Standardize terminology, use either "antibiotic" or "antimicrobial" consistently.

Response: Thank you for your comment. We standardized the word to "antimicrobial" consistently in the manuscript.

Comment 15: Lines 189 and 191: Include the MIC ranges defining high and low resistance rates.

Response: Thank you for your comment. The MIC ranges defining susceptibility and resistance for each antimicrobial have been added in the Results section (Lines 195-201 in the clean version).

Comment 16: Line 263: Rephrase the sentence beginning with "about" for clarity.

Response: Thank you for the suggestion. The section title has been revised from “Genome analysis about strains from the same patient” to “Comparative genomic analysis of strains from the same patient” to improve clarity.

Comment 17: Line 274: Correct "MS148" to "E148."

Response: Thank you for your comment. “MS148” was a typographical mistake and has been corrected to “E148” (Lines 282 in the clean version).

Comment 18: Line 311: Rephrase or remove "it was" for sentence flow.

Response: Thank you for your comment. Sentence rephrased as suggested (Line 316 in the clean version).

Comment 19: Figure S1: Add color coding to enhance clarity, and consider moving this figure into the main manuscript instead of leaving it in the supplementary section.

Response: Thank you for your comment. We have added color coding to highlight cluster-group relationships and moved the figure into the main manuscript as Figure 3 for better visualization. We also modified the figure legend accordingly. (Lines 433-439 in the clean version)

Re: Spectrum02865-25R1 (**Precise Species Identification and Whole-Genome Sequencing Analysis of *Enterobacter cloacae* complex Causing Bloodstream Infections in China**)

Dear Prof. Ying-Chun Xu:

Your manuscript has been accepted, and I am forwarding it to the ASM production staff for publication. Your paper will first be checked to make sure all elements meet the technical requirements. ASM staff will contact you if anything needs to be revised before copyediting and production can begin. Otherwise, you will be notified when your proofs are ready to be viewed.

Sincerely,
Vittal Ponraj
Editor
Microbiology Spectrum

Reviewer #1 (Comments for the Author):

The authors have responded positively and constructively to each of the suggestions and comments of the previous reviews. As such, the manuscript has been improved and the authors' conclusions better supported by the presented data.

Reviewer #2 (Comments for the Author):

No comments